# LASP-2: Rethinking Sequence Parallelism for Linear Attention and its Hybrid

## Abstract

Linear sequence modeling approaches, such as linear attention (Katharopoulos et al., 2020), provide advantages like linear-time training and constant-memory inference over sequence lengths. However, existing sequence parallelism (SP) methods are either not optimized for their right-product-first feature or use ring-style communication as in LASP (Sun et al., 2024), which results in lower computation parallelism, limits their scalability for longer sequences in distributed systems. In this paper, we introduce LASP-2, a new SP approach designed to enhance both communication and computation efficiency in linear (attention) transformer models with very-long input sequences. Compared to LASP, LASP-2 rethinks the minimal communication requirement for SP on linear attention, reorganizes the whole communication-computation order of LASP. In this way, only one single all-gather collective communication is needed on intermediate memory states, whose sizes are independent of the sequence length, leading to significant improvements of both communication and computation parallelism, as well as their overlap. Additionally, we extend LASP-2 to LASP-2H by applying similar communication redesign to standard attention modules, offering an efficient SP solution for hybrid models that combine linear and standard attention layers. Our evaluation on a Linear-Llama3 model, a variant of Llama3 with linear attention replacing standard attention, demonstrates the effectiveness of LASP-2 and LASP-2H. Specifically, LASP-2 achieves throughput improvements of 15.2% over LASP and 36.6% over Ring Attention, with a sequence length of 2048K across 64 GPUs.

## 1 Introduction

Transformer, originally introduced by Vaswani et al. (Vaswani et al., 2017), has become the backbone of modern models across a wide range of domains, including language, vision, audio, video, graphs, and even time-series data. Although the Transformer dates back to 2017, its adaptability and robustness have made it indispensable for a variety of tasks. Central to its success is the self-attention mechanism, which is highly effective for sequence modeling, but has quadratic complexity (w.r.t. sequence length), leading to significant computational costs during training. However, the ability to handle long-context sequences is crucial for large model applications, not only for language tasks but also for multi-modal tasks, where sequences naturally tend to be long (Xue et al., 2024). FlashAttention series (Dao et al., 2022; Dao, 2023; Shah et al., 2024) have provided substantial advancements in scaling attention to handle longer sequences by optimizing the CUDA-level computations for better hardware utilization. However, the theoretical complexity of FlashAttention remains quadratic. Moreover, the need to maintain the KV cache presents further difficulties in managing memory as the sequence length extends (Yang et al., 2024). As a result, long-sequence processing in Transformer models continues to be a complex and resource-intensive problem.

Recently, numerous variations of attention have been proposed, primarily aimed at addressing its quadratic computational and memory complexity, as well as the growing size of the KV cache (Peng et al., 2023; 2024). One promising approach line is linear attention (Katharopoulos et al., 2020), which replaces the exponential kernel in softmax attention with a simpler dot product between key and query vectors. This shift allows linear attention to be structured as a linear recurrent neural network (RNN) with matrix-valued hidden states, thereby eliminating the need for a KV cache. In consequence, it supports constant-memory inference and reduces training complexity from quadratic to linear (Yang et al., 2023). A parallel line of research focuses on State Space Models (SSMs), such

as Mamba (Gu & Dao, 2023) and Mamba 2 (Dao & Gu, 2024), which draw upon concepts from control theory. Both linear attention and SSMs share a common recurrent formulation, expressed as $\mathbf{M}_s = \mathbf{M}_{s-1} + \widehat{\mathbf{M}}_s$, where $\widehat{\mathbf{M}}_s$ represents the incremental memory state of the $s$-th token (Yang et al., 2024). However, despite these advantages, they tend to perform poorly on recall-intensive tasks, such as in-context learning (e.g., five-shot MMLU (Hendrycks et al., 2020), Phone-book lookup (Jelassi et al., 2024), Needle In A Haystack (Briakou et al., 2023)) and long-context reasoning. Empirical research (Lieber et al., 2024; Ren et al., 2024; Waleffe et al., 2024) has shown that models relying solely on linear sequence modeling struggle to excel in these domains. However, a hybrid architecture combining linear sequence modeling layers with standard transformer layers has been demonstrated to significantly enhance model performance on tasks that are recall-intensive.

Sequence Parallelism (SP) techniques (Korthikanti et al., 2022; Jacobs et al., 2023; Liu et al., 2023) are commonly employed to partition long sequences into smaller sub-sequences, allowing them to be processed across multiple GPUs in parallel. Despite the advantages offered by SP for handling large sequences, current SP methods do not fully exploit the right-product-first feature of linear attention, which can lead to inefficiencies in parallelism and communication. LASP (Sun et al., 2024) (referred to as LASP-1) introduced a SP approach specifically tailored for linear attention, that uses a point-to-point (P2P) communication strategy. In this method, intermediate states are transferred across GPUs in a ring-style pattern within the distributed world. However, although such P2P ring-style communication offers certain benefits, part of its computation has to be executed sequentially, which leads low computation parallelism. In addition, too many small P2P operators make the overlapping of communication and computation difficult.

In this paper, we introduce LASP-2 by rethinking the minimal communication requirement involved in SP of linear attention. Specifically, we innovatively reorganize the whole computation and communication procedure with an optimized execution order. In this way, only a single all-gather collective communication is needed in the forward or backward of each iteration. These bring both communication and computation efficiency improvements: 1) the size of the intermediate memory state tensor the all-gather operator works on is independent of the sequence length, making the communication burden insignificant in the context of long sequences. The communication parallelism and accessibility to overlap with computation are notably improved. 2) the refactored computation order improves computation parallelism over multiple devices. Additionally, we separately present LASP-2 with and without masking for autoregressive and bidirectional tasks, respectively, as the presence of a mask matrix significantly impacts the execution mechanism of LASP-2. To extend LASP-2 to hybrid models with both linear and standard attention modules, we introduce LASP-2H. This extension employs the same all-gather-based communication for standard attention layers, with a similar designing philosophy on linear attention. We conduct experiments with up to sequence length of 2048K to verify the efficiency advantages of LASP-2 and LASP-2H.

The main contributions of this paper can be summarized as follows:

- We rethink the communication design for the current SP on linear attention, reorganize its whole communication and computation process with an optimized execution order. This involves using a single all-gather collective communication on intermediate memory states, whose sizes are independent of sequence length. The resulted LASP-2 significantly improves both communication and computation parallelism, as well as their overlap.
- We extend LASP-2 to LASP-2H, offering an efficient SP solution for hybrid models that integrate both linear and standard attention layers, employing an unified all-gather-based communication strategy.
- We construct a series of Linear-Llama3 models, including both purely linear and hybrid versions. Extensive experimental results on these models with up to a sequence length of 2048K, validate the efficiency improvement and performance of LASP-2 and LASP-2H.

## 2 RELATED WORK

### 2.1 LINEAR SEQUENCE MODELING

**Linear Attention.** Vanilla linear attention (Katharopoulos et al., 2020) introduces the use of kernel methods as a replacement for the Softmax attention (Vaswani et al., 2017), thereby reducing the computational complexity to linear in sequence length. Following this, several variants of linear

Table 1: **Notations.** Indices, operations, constants, vectors and matrices used in the paper.

| **Indices** | | **Operations** | |
|---|---|---|---|
| $i$ | Any indices | $\cdot$ (or omitted) | Matrix multiplication |
| $s$ | Index of current token | $\odot$ | Hadamard multiplication |
| $t$ | Index of chunk | **Vectors and Matrices** | |
| **Constants** | | $\mathbf{x}, \mathbf{o} \in \mathbb{R}^{1 \times d}$ | Input and output vectors |
| $d$ | Hidden dimension | $\mathbf{q}, \mathbf{k}, \mathbf{v} \in \mathbb{R}^{1 \times d}$ | Query, key, value vectors |
| $W$ | World size | $\mathbf{X}, \mathbf{O} \in \mathbb{R}^{N \times d}$ | Input and output matrices |
| $N$ | Sequence length | $\mathbf{Q}, \mathbf{K}, \mathbf{V} \in \mathbb{R}^{N \times d}$ | Query, key, value matrices |
| $T$ | Total number of chunks | $\mathbf{M} \in \mathbb{R}^{d \times d}$ | Memory state matrix |
| $C$ | Chunk length | $\mathbf{W}_Q, \mathbf{W}_K, \mathbf{W}_V \in \mathbb{R}^{d \times d}$ | Weight matrices |

attention have been proposed. RetNet (Sun et al., 2023) introduces a retention mechanism that combines recurrence with attention, benefiting from both parallel training and linear inference. Gated Linear Attention (GLA) (Yang et al., 2023) incorporates a data-independent gating mechanism into the linear attention framework, and presents an efficient algorithm for training. TransNormerLLM (Qin et al., 2024a) proposes Lightning Attention, a refined linear attention mechanism that accelerates processing by optimizing IO interactions. Lightning Attention-2 (Qin et al., 2024b) further realizes the theoretical advantages of linear attention by separately handling inter- and intra-block computations. DeltaNet (Schlag et al., 2021) and its parallelized version (Yang et al., 2024) use a delta rule-like update to enhance linear attention performance in long-context scenarios. Finally, Gated Slot Attention (GSA) (Zhang et al., 2024), inspired by GLA, introduces a gated linear attention mechanism with bounded-memory slot control to further improve efficiency.

**State Space Modeling.** The SSM serves as a powerful framework for representing the behavior of sequences within dynamic systems, and it has shown considerable promise in the realm of linear sequence modeling. Mamba (Gu & Dao, 2023) incorporates a mechanism for selecting states, thereby facilitating the scaling of linear sequence lengths. This architecture has been further enhanced in Mamba-2 (Dao & Gu, 2024), where the introduction of the state space duality (SSD) framework optimizes its performance.

**Linear RNN.** Traditional RNNs face significant challenges in handling long-context sequence modeling, primarily due to their inherent sequence dependency during training, which prevents them from fully capitalizing on scaling laws (Sun et al., 2023). To address these limitations, RWKV (Peng et al., 2023; 2024) was introduced as a linear RNN-based large language model that aims to efficiently manage long-term dependencies. Additionally, HGRN (Qin et al., 2024e) highlights the critical role of data-dependent decay mechanisms in enhancing linear RNN performance, demonstrating how adjustments to decay parameters can improve learning in long-context tasks. An enhanced version, HGRN2 (Qin et al., 2024d), expands on this approach by incorporating a state expansion mechanism that utilizes outer product operations, which allows for greater scalability and improved modeling capabilities over extended sequences. Both RWKV and HGRN models seek to overcome the traditional weaknesses of RNNs for efficient long-sequence modeling.

## 2.2 Sequence Parallelism

SP (Li et al., 2022) is a distributed technology designed for training language models more efficiently, which is implemented by dividing a long sequence into multiple shorter subsequences and processing these subsequences in parallel on multiple computing devices. Existing SP methods (Korthikanti et al., 2022; Jacobs et al., 2023) whose parallelism degree cannot exceed the number of attention heads, which limits their scalability. Ring Attention (Liu et al., 2023) is proposed to address high memory cost in long sequence modeling by distributing subsequences across different devices and overlapping the communication of KV blocks. LASP (Sun et al., 2024) proposes a new linear attention-tailored SP strategy based on GPU friendly implementation by utilizing a P2P ring-style communication strategy, but still lacks of optimizations for hybrid model architecture.

## 3 Preliminary

**Notation** In this paper, we ensure the consistent use of notations to enhance clarity. Table 1 provides a list of all the symbols utilized throughout, including indices, constants, vectors, and matrices. Vectors and matrices are represented in boldface. For simplicity, we have omitted the dimensions related to batch size and the number of heads in tensor shapes.

**Linear Attention**   The term "attention" generally refers to a computation that assigns scores to pairs of positions within a sequence, enabling each element to "attend" to others. The most widely used and significant variant of this mechanism is softmax self-attention, which is central to standard transformer models. During training, with an assumption of a single attention head for simplicity, softmax self-attention computes as follows:

$$\mathbf{Q}, \mathbf{K}, \mathbf{V} = \mathbf{X}\mathbf{W}_Q, \mathbf{X}\mathbf{W}_K, \mathbf{X}\mathbf{W}_V, \quad \mathbf{O} = \text{Softmax}(\mathbf{Q}\mathbf{K}^\top)\mathbf{V}. \tag{1}$$

The mechanism of pairwise comparisons (induced by materializing $\mathbf{Q}\mathbf{K}^\top$) leads to the characteristic quadratic training cost of softmax self-attention. Recently, Linear Attention (Katharopoulos et al., 2020) has gained attention as a potential alternative to softmax self-attention, with two key distinctions. First, it removes the $\text{Softmax}(\cdot)$ operation, incorporating it into a kernel feature map. Second, it leverages the associativity of matrix multiplication to reformulate $(\mathbf{Q}\mathbf{K}^\top)\mathbf{V} = \mathbf{Q}(\mathbf{K}^\top\mathbf{V})$. These adjustments reduce both the computation and memory complexity of attention calculation from $O(N^2d)$ to $O(Nd^2)$. This technique is often referred to as the right-product kernel trick because it prioritizes the multiplication on the right side first.

During inference, both softmax self-attention and linear attention handle a single token at each iteration. Given the $s$-th token $\mathbf{x}_s \in \mathbb{R}^{1 \times d}$, softmax self-attention computes requiring the storage of an expanding set of keys $\{\mathbf{k}_1, \cdots, \mathbf{k}_s\}$ and values $\{\mathbf{v}_1, \cdots, \mathbf{v}_s\}$ i.e., the "KV cache", which leads to a significant memory burden when dealing with long input sequences. In linear attention, researchers have experimented with using various nonlinear kernels to replace the $\exp(\cdot)$ function in Eq. 2.

$$\mathbf{q}_s, \mathbf{k}_s, \mathbf{v}_s = \mathbf{x}_s\mathbf{W}_Q, \mathbf{x}_s\mathbf{W}_K, \mathbf{x}_s\mathbf{W}_V, \quad \mathbf{o}_s = \frac{\sum_{i=1}^s \exp(\mathbf{q}_s{\mathbf{k}_i}^\top)\mathbf{v}_i}{\sum_{i=1}^s \exp(\mathbf{q}_s\mathbf{k}_i^\top)}. \tag{2}$$

However, recent studies (Sun et al., 2023; Yang et al., 2023; Qin et al., 2024c) have found that employing a linear kernel (i.e., using the identity function) without a normalizing denominator works effectively in practice. This results in an unnormalized linear attention form as below:

$$\mathbf{o}_s = \sum_{i=1}^s \mathbf{q}_s({\mathbf{k}_i}^\top\mathbf{v}_i) = \mathbf{q}_s\sum_{i=1}^s({\mathbf{k}_i}^\top\mathbf{v}_i) = \mathbf{q}_s\mathbf{M}_s, \tag{3}$$

where $\mathbf{M}_s = \sum_{i=1}^s {\mathbf{k}_i}^\top\mathbf{v}_i$ is the prefix sum of ${\mathbf{k}_i}^\top\mathbf{v}_i$ from $i = 1$ to $s$, which is also known as the memory state in linear attention. This reformulation leads to a recurrent structure for linear attention, resembling the behavior of RNNs as

$$\mathbf{M}_s = \mathbf{M}_{s-1} + \mathbf{k}_s^\top\mathbf{v}_s, \quad \mathbf{o}_s = \mathbf{q}_s\mathbf{M}_s. \tag{4}$$

# 4 METHOD

## 4.1 LASP-2 WITHOUT MASKING

Sequence parallelism methods work by dividing long input sequences into several smaller chunks, which are then distributed across multiple computational devices. Each device independently processes the queries, keys, and values for its assigned chunk in parallel. To complete the attention computation for the entire sequence, necessary communication steps are performed to either gather the results from all devices or exchange information between them. LASP (Sun et al., 2024) was introduced as a sequence parallelism technique designed specifically for the linear attention module.

Let us consider a distributed computing setup where there are $W$ devices, and the input sequence is divided into $T$ chunks, referred to as the sequence parallel size. In the usual case, $T$ is evenly divisible by $W$, and we often assume $W = T$. It means each chunk is assigned to a single device, ensuring that every chunk is processed in parallel across the distributed system. This scenario exemplifies pure sequence parallelism. Additionally, in Sec.A.4.1, we will explore cases where $W \neq T$, representing a hybrid approach that combines sequence parallelism with data parallelism.

In LASP-2, the input sequence $\mathbf{X}$ is divided into $T$ smaller chunks, represented as $[\mathbf{X}_t]_1^T$, and each chunk is distributed across the devices in the distributed system. For each chunk $\mathbf{X}_t$, its corresponding query, key, value, and the linear attention memory state can be computed in parallel across all chunks. This parallel computation is carried out as follows:

$$\mathbf{Q}_t, \mathbf{K}_t, \mathbf{V}_t = \mathbf{X}_t\mathbf{W}_Q, \mathbf{X}_t\mathbf{W}_K, \mathbf{X}_t\mathbf{W}_V, \quad \mathbf{M}_t = \mathbf{K}_t^\top\mathbf{V}_t. \tag{5}$$

**Algorithm 1** LASP-2 w/o Masking

1: **Input:** input sequence $\mathbf{X}$, distributed world size $W$, sequence parallel size $T = W$.
2: Distribute $\mathbf{X} = [\mathbf{X}_t]_1^T$.
3: **for** chunk $t \in \{1, \cdots, T\}$ on ranks $\{1, \cdots, W\}$ **in parallel do**
4:    Calculate $\mathbf{Q}_t = \mathbf{X}_t \mathbf{W}_Q$, $\mathbf{K}_t = \mathbf{X}_t \mathbf{W}_K$, $\mathbf{V}_t = \mathbf{X}_t \mathbf{W}_V$.
5:    Compute $\mathbf{M}_t = \mathbf{K}_t^\top \mathbf{V}_t$.
6:    Communicate

   $[\mathbf{M}_t]_1^T = \texttt{AllGather}([\mathbf{M}_t]_1^T).$

7:    Compute $\mathbf{M}_{1:T} = \texttt{Sum}([\mathbf{M}_t]_1^T)$.
8:    Compute $\mathbf{O}_t = \mathbf{Q}_t \mathbf{M}_{1:T}$.
9: **end for**
10: return $\mathbf{O} = [\mathbf{O}_t]_1^T$.

**Algorithm 2** LASP-2 w/ Masking

1: **Input:** input sequence $\mathbf{X}$, distributed world size $W$, sequence parallel size $T = W$.
2: Distribute $\mathbf{X} = [\mathbf{X}_t]_1^T$.
3: Initialize mask matrix $\boldsymbol{\Psi}$, where $\boldsymbol{\Psi}_{ij} = 1$ if $i \geq j$ and $\boldsymbol{\Psi}_{ij} = -\infty$ if $i < j$.
4: **for** chunk $t \in \{1, \cdots, T\}$ on ranks $\{1, \cdots, W\}$ **in parallel do**
5:    Calculate $\mathbf{Q}_t = \mathbf{X}_t \mathbf{W}_Q$, $\mathbf{K}_t = \mathbf{X}_t \mathbf{W}_K$, $\mathbf{V}_t = \mathbf{X}_t \mathbf{W}_V$.
6:    Compute $\mathbf{M}_t = (\mathbf{K}_t)^\top \mathbf{V}_t$.
7:    Communicate $[\mathbf{M}_t]_1^T = \texttt{AllGather}([\mathbf{M}_t]_1^T)$.
8:    Compute $\mathbf{O}_{t,\text{intra}} = [(\mathbf{Q}_t \mathbf{K}_t^\top) \odot \boldsymbol{\Psi}]\mathbf{V}_t$.
9:    Compute prefix sum $\mathbf{M}_{1:t-1} = \texttt{PrefixSum}([\mathbf{M}_t]_1^{t-1})$.
10:    Compute $\mathbf{O}_{t,\text{inter}} = \mathbf{Q}_t \mathbf{M}_{1:t-1}$.
11:    Compute $\mathbf{O}_t = \mathbf{O}_{t,\text{intra}} + \mathbf{O}_{t,\text{inter}}$.
12: **end for**
13: return $\mathbf{O} = [\mathbf{O}_t]_1^T$.

By performing this concurrent computation for each chunk, LASP-2 efficiently handles long input sequences in a distributed setting. The query $\mathbf{Q}_t$, key $\mathbf{K}_t$, value $\mathbf{V}_t$, and the memory state $\mathbf{M}_t$ are calculated individually for every chunk of the sequence, ensuring that no single device is overburdened with processing the entire sequence at once. This distributed approach facilitates better memory management and computational efficiency, especially when dealing with extremely long sequences. Thus, LASP-2 leverages the power of sequence partitioning to optimize the calculation of linear attention in a distributed framework.

Notably, in LASP-2, only a single all-gather collective communication operation is required during the forward pass. This all-gather operation acts on the memory states $[\mathbf{M}_t]_1^T$ associated with each sequence chunk, ensuring that every device in the system has access to the complete set of memory states $[\mathbf{M}_t]_1^T$. Once the memory states from all chunks have been gathered, they are concurrently accumulated on all devices to compute the memory state corresponding to the entire input sequence. This process is expressed as follows:

$$[\mathbf{M}_t]_1^T = \texttt{AllGather}([\mathbf{M}_t]_1^T), \quad \mathbf{M}_{1:T} = \texttt{Sum}([\mathbf{M}_t]_1^T). \tag{6}$$

Finally, the linear attention output corresponding to the local query $\mathbf{Q}_t$ can be computed as:

$$\mathbf{O}_t = \mathbf{Q}_t \mathbf{M}_{1:T}.$$

Importantly, the accumulation step $\texttt{Sum}([\mathbf{M}_t]_1^T)$ can be efficiently performed in a recursive manner, by adding each memory state sequentially as $\mathbf{M}_{1:t-1} + \mathbf{M}_t$. This eliminates the need to repeatedly calculate the sum of the memory states from earlier chunks, improving the efficiency of the computation. To further optimize performance, we cache the accumulated result $\mathbf{M}_{1:T}$ in high-bandwidth memory (HBM). This caching strategy speeds up the backward pass by avoiding redundant recalculations of $\mathbf{M}_{1:T}$, which is necessary for computing gradients. This approach is akin to the concept of activation checkpointing, where intermediate activations are saved to avoid recomputation.

It is important to point out that each memory state $\mathbf{M}_t$ has dimensions of $d \times d$, which means the communication cost for the all-gather operation is independent of the sequence or chunk length. Instead, the cost scales linearly with the number of devices involved in the SP communication group. For clarity, we provide a summary of the LASP-2 method, without considering the attention mask, in Algorithm 1. During the backward pass, a similar all-gather communication operation on the gradients of memory states $d\mathbf{M}_t$ is required. The details of this backward pass without masking, can be found in Algorithm 3 in Appendix A.1 for further reference.

## 4.2 LASP-2 WITH MASKING

In autoregressive tasks, the mask matrix $\boldsymbol{\Psi} \in \{-\infty, 1\}^{N \times N}$ is typically a lower triangular matrix, where $\boldsymbol{\Psi}_{ij} = 1$ for $i \geq j$ and $\boldsymbol{\Psi}_{ij} = -\infty$ when $i < j$. This structure enforces a causal constraint during computation. Specifically, when calculating $\mathbf{O} = \texttt{Softmax}(\mathbf{Q}\mathbf{K}^\top \odot \boldsymbol{\Psi})\mathbf{V}$, it becomes impossible to leverage the associative property of matrix multiplication to reduce the computational complexity from quadratic to linear in a parallel form.

To address this challenge in linear attention with a causal mask, we adopt the approach of computation decomposition, as proposed in earlier work (Yang et al., 2023; Sun et al., 2024). Figure 1 provides an illustration that highlights the difference between intra-chunk and inter-chunk computations in linear attention. Inter-chunk calculations, which have no dependencies on other chunks across devices, can be treated as if they have no causal mask. As a result, these computations can be parallelized across all devices in the distributed setup. In contrast, intra-chunk calculations account for the influence of previous chunks (1 to $(t-1)$) on the $t$-th chunk. These intra-chunk operations are affected by the mask matrix, and therefore, require specialized handling to respect the causal constraints.

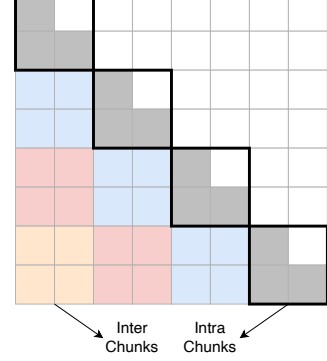

Figure 1: **Computation Decomposition in LASP-2 with masking.** Colored chunks represent inter-chunks.

For linear attention computation on intra-chunks, given the query, key, and value matrices $\mathbf{Q}_t$, $\mathbf{K}_t$, and $\mathbf{V}_t$ corresponding to the chunk $\mathbf{X}_t$, the output is computed as

$$\mathbf{O}_{t,\text{intra}} = [(\mathbf{Q}_t\mathbf{K}_t^\top) \odot \boldsymbol{\Psi}]\mathbf{V}_t, \tag{7}$$

This formulation adheres to the standard left-product matrix multiplication. Although the computation can be executed in parallel across devices, it retains the quadratic complexity commonly associated with traditional attention mechanisms during training. This limitation arises from the element-wise masking operation ($\odot \boldsymbol{\Psi}$), which enforces causal constraints within the chunk, preventing the use of optimizations that would reduce the computational cost to linear.

For linear attention computation across inter-chunks, we follow a similar approach as the procedure outlined for LASP-2 without masking. First, the memory states for each chunk are computed concurrently across different devices as $\mathbf{M}_t = \mathbf{K}_t^\top\mathbf{V}_t$. These memory states, corresponding to each chunk, are initially distributed across separate devices. To synchronize the results, an `AllGather` collective communication operation is performed. This step ensures that all devices hold the memory states for all chunks, enabling further parallel processing. Once the memory states have been gathered, we proceed with a concurrent `PrefixSum` operation across all devices. This operation accumulates the memory states from the 1st chunk up to the $(t-1)$-th chunk, effectively building the necessary intermediate states. This can be expressed as:

$$[\mathbf{M}_t]_1^T = \texttt{AllGather}([\mathbf{M}_t]_1^T), \quad \mathbf{M}_{1:t-1} = \texttt{PrefixSum}([\mathbf{M}_t]_1^{t-1}). \tag{8}$$

The `PrefixSum` operation can be optimized by implementing it recursively, utilizing cached memory states stored on the HBM. Specifically, the accumulation of memory states is computed as:

$$\mathbf{M}_{1:t-1} = \mathbf{M}_{1:t-2} + \mathbf{M}_{t-1}. \tag{9}$$

By caching $\mathbf{M}_{1:t-1}$, the backward pass computation is facilitated since this cached value is a necessary activation for gradient calculations. This approach not only speeds up the backward pass but also reduces the computational load, as the cached memory state eliminates the need for repeated re-computation.

Following the calculation of the memory states, the outputs corresponding to the inter-chunks and the final output for the $t$-th token can be derived with ease. The overall output for the $t$-th token is obtained by summing both the intra-chunk and inter-chunk outputs.

$$\mathbf{O}_{t,\text{inter}} = \mathbf{Q}_t\mathbf{M}_{1:t-1}, \quad \mathbf{O}_t = \mathbf{O}_{t,\text{intra}} + \mathbf{O}_{t,\text{inter}}. \tag{10}$$

We provide the complete algorithm for LASP-2 with masking in Algorithm 2, and its backward pass in Algorithm 4 in Appendix A.1. It is important to note that the communication operation in line 7, along with the computation of $\mathbf{O}_{t,\text{intra}}$ in line 8, can be overlapped by executing them on separate threads. This concurrent execution helps improve overall efficiency, as it allows for the overlap of communication and computation, effectively reducing idle time during the process.

### 4.3 LASP-1 vs LASP-2

LASP-2, as well as its previous version LASP-1, both aim on efficient SP on linear attention. Although, in theory, LASP-1 and LASP-2 share similarity on communicating the KV activation ($d \times d$), whose

size is independent of the sequence or chunk length. They have fundamental distinctions where the key differences lie in their communication manners and the computational order reorganization, as elaborated as below:

- LASP-1 utilizes a ring-style P2P communication, which needs to launch many `send` & `receive` operators between devices, to sequentially transfer the KV activation one-by-one among the devices. This makes the communication process relatively slow and hard to adequately overlap with intra-chunk computations.
- While LASP-2 uses a single `AllGather` collective communication operator to exchange KV activation concurrently among all decices. This offers practical advantages: (1) Only one well-optimized collective communication operator needs to be launched, and the exchange of KV activation on all devices can be finished concurrently all at once; (2) the collective communication can be more easily overlapped with computations. Like in LASP-2 with masking, the `AllGather` communication is able to overlap with the intra-chunk output computations. And, in addition, LASP-2 reorganizes the whole computation order to make the `AllGather` based communication strategy feasible and efficiency.

We also write down the Algorithms of LASP-1 (with and without masking) in identical mathematical symbols in Appendix A.2 for convenience to compare with LASP-2 on their algorithmic differences.

## 4.4 THEORETICAL COST ANALYSIS

For better understanding the superiorities of LASP-2, we provide a theoretical cost analysis of both LASP-1 and LASP-2. We consider the pure SP scenario, i.e., the distributed world size is $W$, and an input sequence with a length of $N$ is partitioned into $T = W$ chunks, thus all devices in this world need to involve into the communication. Below $B$ denotes batch size, $H$ represents number of heads.

**Communication traffic in each communication step: LASP-1:** $BHd^2$**, LASP-2:** $BHd^2$**.** This is because both LASP-1 and LASP-2 transfer linear attention memory states (not keys and values) among devices. The memory state corresponding to each chunk (located at each device) has a tensor shape of [B, H, d, d]. Thus in each communication step, their communication traffic are both $BHd^2$.

For a practical Linear-Llama3-1B model with $B = 16$, $H = 16$ and $d = 2048$, each memory state will has $BHd^2 \approx 1.07$ billion parameters, which takes around $2.14GB$ memory in FP16. For a Linear-Llama3-8B model with $B = 16$, $H = 32$ and $d = 4096$, each memory state will has $BHd^2 \approx 8.59$ billion parameters, which takes around $17.18GB$ memory in FP16.

**Number of communication steps in each iteration: LASP-1:** $2(W - 1)$**, LASP-2:** $2$**.** This depends on the different communication manners of these two algorithms. During the forward of an iteration, LASP-2 launches a single all-gather operation to gather all memory states $\mathbf{M}_t$ to all devices, i.e., $[\mathbf{M}_t]_1^T = \texttt{AllGather}([\mathbf{M}_t]_1^T)$. This collective operation is concurrently executed on all devices. While in backward, another all-gather is performed on the gradients of $\mathbf{M}_t$, i.e., $[\mathbf{dM}_t]_1^T = \texttt{AllGather}([\mathbf{dM}_t]_1^T)$. Thus in each iteration, LASP-2 has 2 communication steps. While LASP-1 uses a pair of `send` & `receive` operation to sequentially exchange the memory state from one device to another device. During forward, device $i$ sends its memory state to device $i + 1$, and device $i + 1$ receives the memory state from device $i$, and so on. Computations of $\mathbf{O}_{t,\text{inter}}$, $\mathbf{O}_t$ and updates of $\mathbf{M}_t$ are followed behind each `receive` operation on that device. Thus in the process of forward, LASP-1 has $W - 1$ communication steps. In the backward, this process is repeated reversely from the last device to device 0. Thus in each iteration, LASP-1 have totally $2(W - 1)$ communication steps.

Given that both LASP-1 and LASP-2 perform a total of $I$ iterations, their communication traffic models can be expressed as follows: LASP-1: $2(W - 1)IBHd^2$ and LASP-2: $2IBHd^2$. Ideally, LASP-2's communication traffic would be reduced by a factor of $W - 1$ compared to LASP-1. However, the actual communication cost depends on factors like communication bandwidth, which is typically faster within nodes and slower across nodes. As a result, LASP-2's benefits become more evident in clusters with slower interconnects, and vice versa. It is important to note that this cost model only accounts for communication, excluding computation or data loading. In practice, communication represents a smaller portion of the total cost, so the overall training speedup achieved by LASP-2 is less than $W - 1$ times. LASP-2 performs best in scenarios involving long sequences, large clusters, slow communication links, and efficient data loading and computation.

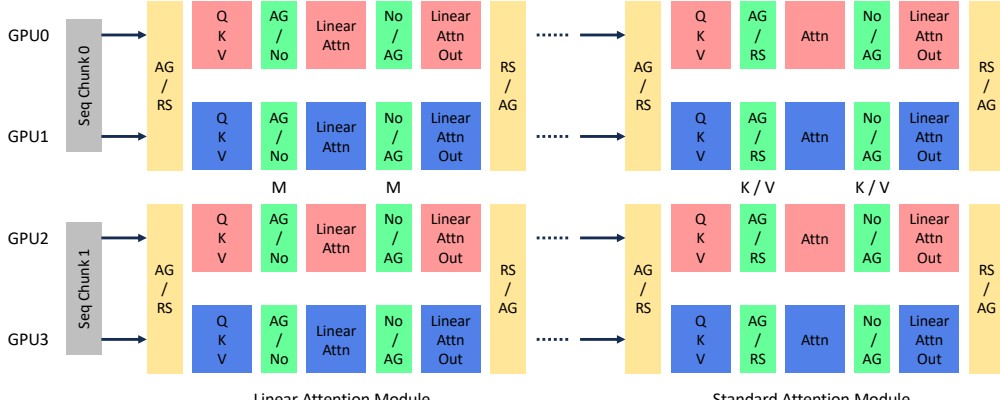

Figure 2: **Visualization of LASP-2H on Linear Attention and Standard Attention hybrid model.**
We exemplify LASP-2H on the hybrid layers of linear attention and standard attention modules with
both TP and SP (both have a dimension of 2). The communication operations colored in yellow and
green are for TP and SP, respectively. AG/RS: all-gather in forward and reduce-scatter in backward,
RS/AG: reduce-scatter in forward and all-gather in backward, AG/No: all-gather in forward and no-op
in backward, No/AG: no-op in forward and all-gather in backward. Note that the SP communication
operations for linear attention operate on the memory state $\mathbf{M}_t \in \mathbb{R}^{d \times d}$, while for standard attention,
they operate on states $\mathbf{K}_t, \mathbf{V}_t \in \mathbb{R}^{C \times d}$.

### 4.5 HYBRID MODEL SEQUENCE PARALLELISM

The hybrid model, which combines linear transformer layers with standard transformer layers that
utilize softmax self-attention, has been demonstrated to effectively enhance long-context capabilities,
particularly in tasks such as recall and retrieval. To optimize SP in such hybrid models, we propose
an extended version of LASP-2, referred to as LASP-2H. This approach introduces a comprehensive
solution by incorporating SP into both the linear attention and standard attention modules. The
structure of LASP-2H is illustrated in Fig. 2.

**On Linear Attention Module.** As outlined in Algorithm 1 and Algorithm 2, LASP-2H handles linear
attention modules by performing a single all-gather communication operation on the memory state
$\mathbf{M}_t \in \mathbb{R}^{d \times d}$. The communication complexity remains independent of both sequence or chunk length,
and only scales linearly with the SP size $T$, making this method efficient in distributed clusters.

**On Standard Attention Module.** Context Parallelism (CP) is a SP technique in Megatron-LM that
divides network inputs and all activations along the sequence dimension. This approach is specifically
tailored for standard softmax attention. While traditional CP implementations in Megatron-LM
rely on overlapping communication and computation in a ring-like structure (Liu et al., 2023), our
LASP-2H adopts a different method, following the best practice in Llama3 (Dubey et al., 2024).
Instead of the ring-style strategy, LASP-2H employs AllGather-based communication on standard
attention, where the $\mathbf{K}_t$ and $\mathbf{V}_t$ tensors are first gathered across devices, after which the attention
output is computed locally for the $\mathbf{Q}_t$ tensor chunk. Although the all-gather communication has a
higher latency compared to ring-based methods, it provides greater ease and flexibility in handling
various types of attention masks, such as document-level masks. This flexibility is particularly
beneficial in scenarios where different attention patterns are needed. Additionally, the all-gather
latency is minimized because the $\mathbf{K}_t$ and $\mathbf{V}_t$ tensors are significantly smaller than the $\mathbf{Q}_t$ tensor,
especially when using Grouped Query Attention (GQA) (Ainslie et al., 2023). As a result, the time
complexity of computing the attention output far exceeds the complexity of all-gather operation. We
present the description of AllGather-based Context Parallelism in Algorithm 7 in Appendix A.3.

## 5 EXPERIMENTS

We conducted an empirical evaluation of LASP-2 by applying it to a model based on Llama3 (Dubey
et al., 2024). We replaced the standard softmax attention with various linear attention modules,
including the original basic linear attention (Katharopoulos et al., 2020), Lightning Attention (Qin
et al., 2024b), Retention (Sun et al., 2023), Gated Linear Attention (GLA) (Yang et al., 2023),
Based (Arora et al., 2024), and Rebased (Aksenov et al., 2024). This modified model, termed Linear-

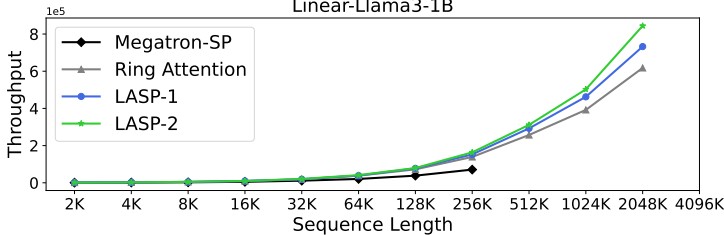

Figure 3: **Throughput Comparison (tokens/s).** Experiments were carried out on a pure Linear-Llama3-1B model, utilizing the basic linear attention module. A total of 64 A100 GPUs were employed, and the SP size $T$ was also set to 64. To accommodate very-long sequence lengths, such as 2048K, the batch size was kept fixed at 1 throughout this experiment.

Llama3, comprises 16 (linear transformer) layers, with a total of 1B parameters. Additionally, we created a hybrid model by retaining transformer layers with standard softmax attention at every fourth layer of Linear-Llama3, forming a $1/4$ hybrid architecture. All experiments were conducted on the SlimPajama dataset (Soboleva et al., 2023), utilizing the Llama3 tokenizer (Dubey et al., 2024). The full dataset contains 627B tokens, but for our experiments, we used a 50B tokens subset derived from the first chunk of the training split. The experiments were performed using GPT-style autoregressive language modeling tasks with attention masks, as this setup mirrors many practical scenarios where such tasks are commonly applied. It is important to note that the primary focus of these experiments is to assess the training efficiency of LASP-2 when handling very-long input sequences. Training a large language model with optimal long-context capabilities falls outside the scope of this study.

### 5.1 EXPERIMENTAL SETUP

**Hardware and Software.** Our experiments were conducted on a configuration of up to 16 DGX-A100 servers, each equipped with 8 A100 GPUs. The GPUs are connected through NVSwitch, offering an inter-GPU bandwidth of 600 GBps. The experiments were implemented using PyTorch 2.3.1, with support from CUDA 12.1, cuDNN 8.9.2, and NCCL 2.20.5. The algorithm was developed on top of NVIDIA's Megatron-Core 0.9.0 (Shoeybi et al., 2019). We use Triton 2.3.1 (Tillet et al., 2019) to accelerate the linear attention computation on GPU, and take FlashAttention-2 (Dao, 2023) as the standard attention implementation. When implement other SP methods (e.g., Ring Attentoin, Megatron-SP) on linear attention instances for the purpose of comparison, we do not incorporate the right-product kernel trick. We maintain the use of each method's original communication primitives and computational manners as they originally proposed for standard attention.

**Hyperparameters.** For training the Linear-Llama3 model, we employed a cosine learning rate schedule with a linear warm-up phase. The minimum learning rate was set to $1e^{-6}$. We applied gradient clipping with a value of 1.0 and weight decay at a rate of 0.1. The Adam optimizer (Kingma & Ba, 2014) was used, configured with $\beta_1 = 0.9$ and $\beta_2 = 0.95$. Additionally, the dropout rate in both attention and hidden layers was set to 0.

### 5.2 SPEED

To assess the speed performance of our proposed LASP-2, we conducted a comparison against existing SP methods, including Megatron-SP (Korthikanti et al., 2022), Ring Attention (Liu et al., 2023), and LASP-1 (Sun et al., 2024). As depicted in Fig. 3, LASP-2 demonstrated superior throughput, particularly when sequence lengths exceeded 64K. This performance advantage became increasingly prominent as sequence lengths grew longer. Specifically, at a sequence length of 512K, LASP-2 outperformed Ring Attention by 17.8% and surpassed LASP-1 by 7.3%. This advantage became even more pronounced at a sequence length of 2048K, where LASP-2 achieved throughput gains of 36.6% over Ring Attention and 15.2% over LASP-1.

### 5.3 SCALABILITY

We assessed the scalability of LASP-2 in terms of both GPU memory usage and throughput by adjusting the sequence length and the number of GPUs. The results were displayed in Figure 4. LASP-2 demonstrated the ability to scale linearly with the input sequence length by increasing the number of GPUs. For instance, while maintaining the same memory cost per GPU, using 8 GPUs allowed training on sequences up to 128K in length, whereas 128 GPUs (16 × 8 GPUs) enabled

Figure 4: **Scalability Results.** Experiments were conducted on a pure Linear-Llama3-1B model using the basic linear attention module. SP size $T$ was always set equally to the total number of GPUs. Batch size was fixed as 1 to accommodate very-long sequence lengths, e.g., 2048K. The sign "×" with a dotted line represents occurring an Out of Memory (OOM).

Table 2: **Convergence Performance Results.** All experiments used 8 A100 GPUs, sequence length of 16K, and batch size of 8, trained on 50B tokens from the SlimPajama corpus.

| Model | SP Method | Attention Module | Pure Model | | 1/4 **Hybrid Model** | |
|---|---|---|---|---|---|---|
| | | | **Thpt** | **Loss** | **Thpt** | **Loss** |
| Llama3 | Ring Attention | Standard Attention | 16549.5 | 2.759 | \ | \ |
| Linear-Llama3 | LASP-2(H) | Basic Linear Attention | 17834.3 | 2.892 | 17394.7 | 2.824 |
| | | Lightning Attention | 17926.1 | 2.862 | 17384.2 | 2.758 |
| | | Retention | 17859.6 | 2.867 | 17352.5 | 2.759 |
| | | GLA | 17785.3 | 2.845 | 17273.2 | 2.754 |
| | | Based | 17946.1 | 2.754 | 17462.5 | 2.751 |
| | | Rebased | 17896.2 | 2.845 | 17284.5 | 2.787 |

training on sequences as long as 2048K (16 × 128K). Additionally, we observed that increasing both the sequence length and the number of GPUs results in higher throughput, indicating improved communication efficiency and linear scalability in LASP-2. More detailed quantitative scalability outcomes were provided in Table 6 in Appendix A.5.

## 5.4 PERFORMANCE

We conducted additional experiments to assess the pretraining convergence performance of LASP-2 on Llama-3 with various attention modules, including standard softmax attention, basic linear attention, Lightning Attention, Retention, GLA, Based, Rebased, and their 1/4 hybrid models. All experiments were performed on the SlimPajama corpus (Soboleva et al., 2023), using 50B tokens, a sequence length of 16K, and a global batch size of 8, using 8 A100 GPUs. The results, as shown in Table 2, indicated that for pure Linear-Llama3 models with different linear attention modules, LASP-2 achieved comparable, though slightly higher, loss values while maintaining superior throughput. On the 1/4 hybrid Linear-Llama3 model, the loss results were generally better than those of the pure linear models, with Lightning Attention, Retention, and GLA even attaining equivalent or lower loss values compared to the baseline. The Based attention module shows strong throughput and loss performance, since its original design uses a mix of (Taylor) linear attention and sliding window attention. The 1/4 hybrid model striked a balance between throughput and convergence performance, performing competitively when compared to both the baseline Llama3 and its pure linear version.

## 6 CONCLUSION

This paper presents LASP-2, a new SP method that addresses the inefficiencies of existing SP approaches for linear sequence modeling. By reorganizing the whole algorithm execution order and leveraging an all-gather communication strategy, LASP-2 enhances both communication and computation parallelism, and enables easier communication-computation overlapping, comparing with LASP-1. Our results demonstrate that LASP-2 offers significant improvements in throughput and scalability, especially in the context of very-long sequence length. Furthermore, the extension to LASP-2H enables efficient SP in hybrid models that integrate both linear and standard attention modules, both utilize an unified all-gather-based communication primitive. Experimental evaluations on the Linear-Llama3 models validate these advancements, with LASP-2 outperforming previous methods like LASP-1 and Ring Attention by substantial margins, particularly at extreme sequence lengths. These findings confirm the practical utility of LASP-2 for large-scale distributed systems, making it a promising approach for future applications in long-sequence linear transformer models.

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

# A  APPENDIX

## A.1  LASP-2 ALGORITHMS (BACKWARD PASS)

See Algorithm 3 and Algorithm 4.

---

**Algorithm 3** LASP-2 w/o Masking (Backward Pass)

---

1: **Input:** distributed world size $W$, sequence parallel size $T = W$, $\mathbf{Q}_t, \mathbf{K}_t, \mathbf{V}_t, \mathbf{O}_t, \mathbf{dO}_t \in \mathbb{R}^{C \times d}$ for chunk $t \in \{1, \cdots, T\}$.
2: **for** chunk $t \in \{1, \cdots, T\}$ on ranks $\{1, \cdots, W\}$ **in parallel do**
3:     Compute $\mathbf{dM}_t = (\mathbf{Q}_t)^\top \mathbf{dO}_t$.
4:     Communicate $[\mathbf{dM}]_1^T = \texttt{AllGather}([\mathbf{dM}]_1^T)$.
5:     Compute $\mathbf{dM}_{1:T} = \texttt{Sum}([\mathbf{dM}]_{t+1}^T)$.
6:     Compute $\mathbf{dQ}_t = \mathbf{dO}_t \mathbf{M}_{1:T}^\top$.
7:     Compute $\mathbf{dK}_t = \mathbf{V}_t \mathbf{dM}_{1:T}^\top$.
8:     Compute $\mathbf{dV}_t = \mathbf{K}_t \mathbf{dM}_{1:T}$.
9: **end for**
10: return $\mathbf{dQ} = [\mathbf{dQ}_t]_1^T$, $\mathbf{dK} = [\mathbf{dK}_t]_1^T$, $\mathbf{dV} = [\mathbf{dV}_t]_1^T$.

---

**Algorithm 4** LASP-2 w/ Masking (Backward Pass)

---

1: **Input:** distributed world size $W$, sequence parallel size $T = W$, $\mathbf{Q}_t, \mathbf{K}_t, \mathbf{V}_t, \mathbf{O}_t, \mathbf{dO}_t \in \mathbb{R}^{C \times d}$ for chunk $t \in \{1, \cdots, T\}$.
2: **for** chunk $t \in \{1, \cdots, T\}$ on ranks $\{1, \cdots, W\}$ **in parallel do**
3:     Compute $\mathbf{dM}_t = (\mathbf{Q}_t)^\top \mathbf{dO}_t$.
4:     Communicate $[\mathbf{dM}]_1^T = \texttt{AllGather}([\mathbf{dM}]_1^T)$.
5:     Compute $\mathbf{dQ}_{t,\text{intra}} = [(\mathbf{dO}_t \mathbf{V}_t^\top) \odot \mathbf{\Psi}] \mathbf{K}_t$.
6:     Compute $\mathbf{dK}_{t,\text{intra}} = [(\mathbf{dO}_t \mathbf{V}_t^\top) \odot \mathbf{\Psi}]^\top \mathbf{Q}_t$.
7:     Compute $\mathbf{dV}_{t,\text{intra}} = [(\mathbf{Q}_t \mathbf{K}_t^\top) \odot \mathbf{\Psi}]^\top \mathbf{dO}_t$.
8:     Compute $\mathbf{dQ}_{t,\text{inter}} = \mathbf{dO}_t \mathbf{M}_{1:t-1}^\top$.
9:     Compute suffix sum $\mathbf{dM}_{t+1:T} = \texttt{SuffixSum}([\mathbf{dM}]_{t+1}^T)$.
10:     Compute $\mathbf{dK}_{t,\text{inter}} = \mathbf{V}_t \mathbf{dM}_{t+1:T}^\top$.
11:     Compute $\mathbf{dV}_{t,\text{inter}} = \mathbf{K}_t \mathbf{dM}_{t+1:T}$.
12:     Combine intra- and inter-chunk parts of $\mathbf{dQ}_t, \mathbf{dK}_t, \mathbf{dV}_t$

$$\mathbf{dQ}_t = \mathbf{dQ}_{t,\text{intra}} + \mathbf{dQ}_{t,\text{inter}}, \quad \mathbf{dK}_t = \mathbf{dK}_{t,\text{intra}} + \mathbf{dK}_{t,\text{inter}}, \quad \mathbf{dV}_t = \mathbf{dV}_{t,\text{intra}} + \mathbf{dV}_{t,\text{inter}}.$$

13: **end for**
14: return $\mathbf{dQ} = [\mathbf{dQ}_t]_1^T$, $\mathbf{dK} = [\mathbf{dK}_t]_1^T$, $\mathbf{dV} = [\mathbf{dV}_t]_1^T$.

---

## A.2  LASP-1 ALGORITHMS

See Algorithm 5 and Algorithm 6.

## A.3  ALLGATHER-BASED CONTEXT PARALLELISM

See Algorithm 7.

## A.4  COMPATIBILITY

### A.4.1  HYBRID PARALLELISM

LASP-2 enables the selection of a sequence parallel size that is smaller and divisible by the distributed world size. This setup splits the input data along both the batch and sequence dimensions, a parallelization strategy known as data-sequence hybrid parallelism. The ZeRO-series optimizers (Rajbhandari et al., 2020) and FSDP (Zhao et al., 2023) offer methods for distributing model states such as optimizer states, gradients, and model parameters across all GPUs in the distributed system. As these techniques are variants of data parallelism, they integrate seamlessly with LASP. Their primary

---

**Algorithm 5** LASP-1 w/o Masking Algorithm

---

1: **Input:** input sequence $\mathbf{X}$, distributed world size $W$, sequence parallel size $T = W$.
2: Distribute input $\mathbf{X} = [\mathbf{X}_t]_1^T$.
3: **for** chunk $t \in \{1, \cdots, T\}$ at rank $i \in \{1, \cdots, W\}$ **in parallel do**
4:     Compute $\mathbf{Q}_t = \mathbf{X}_t \mathbf{W}_Q$, $\mathbf{K}_t = \mathbf{X}_t \mathbf{W}_K$, $\mathbf{V}_t = \mathbf{X}_t \mathbf{W}_V$.
5:     Compute $\mathbf{M}_t = \mathbf{K}_t^\top \mathbf{V}_t$.
6: **end for**
7: **for** chunk $t \in \{1, \cdots, T\}$ at rank $i \in \{1, \cdots, W\}$ **sequentially do**
8:     `Recv` activation $\mathbf{M}_{t-1}$ from rank $(i-1)$. Save $\mathbf{M}_{t-1}$ in memory for backward computation.
9:     Compute $\mathbf{O}_t = \mathbf{Q}_t \mathbf{M}_{t-1}$.
10:    Update $\mathbf{M}_t = \mathbf{M}_{t-1} + \mathbf{K}_t^\top \mathbf{V}_t$.
11:    `Send` activation $\mathbf{M}_t$ to rank $(i+1)$.
12: **end for**
13: **return** $\mathbf{O} = [\mathbf{O}_t]$ with $t \in \{1, \cdots, T\}$.

---

**Algorithm 6** LASP-1 w/ Masking Algorithm

---

1: **Input:** input sequence $\mathbf{X}$, distributed world size $W$, sequence parallel size $T = W$.
2: Distribute input $\mathbf{X} = [\mathbf{X}_t]_1^T$.
3: Initialize mask matrix $\mathbf{\Psi}$, where $\mathbf{\Psi}_{ij} = 1$ if $i \geq j$, and $\mathbf{\Psi}_{ij} = -\infty$ if $i < j$.
4: **for** chunk $t \in \{1, \cdots, T\}$ at rank $i \in \{1, \cdots, W\}$ **in parallel do**
5:     Compute $\mathbf{Q}_t = \mathbf{X}_t \mathbf{W}_Q$, $\mathbf{K}_t = \mathbf{X}_t \mathbf{W}_K$, $\mathbf{V}_t = \mathbf{X}_t \mathbf{W}_V$.
6:     Compute $\mathbf{M}_t = (\mathbf{K}_t)^\top \mathbf{V}_t$.
7:     Compute $\mathbf{O}_{t,\mathrm{intra}} = [(\mathbf{Q}_t \mathbf{K}_t^\top) \odot \mathbf{\Psi}] \mathbf{V}_t$.
8: **end for**
9: **for** chunk $t \in \{1, \cdots, T\}$ at rank $i \in \{1, \cdots, W\}$ **sequentially do**
10:    `Recv` activation $\mathbf{M}_{t-1}$ from rank $(i-1)$. Save $\mathbf{M}_{t-1}$ in memory for backward computation.
11:    Compute $\mathbf{O}_{t,\mathrm{inter}} = \mathbf{Q}_t \mathbf{M}_{t-1}$.
12:    Compute $\mathbf{O}_t = \mathbf{O}_{t,\mathrm{intra}} + \mathbf{O}_{t,\mathrm{inter}}$.
13:    Update $\mathbf{M}_t = \mathbf{M}_{t-1} + \mathbf{K}_t^\top \mathbf{V}_t$.
14:    `Send` activation $\mathbf{M}_t$ to rank $(i+1)$.
15: **end for**
16: **return** $\mathbf{O} = [\mathbf{O}_t]$ with $t \in \{1, \cdots, T\}$.

---

objective of minimizing the memory footprint of model states complements LASP-2's specific focus on reducing activation memory on each GPU, making the training of large-scale models that handle long sequence lengths significantly more manageable.

LASP-2 also offers support for both tensor parallelism (TP) and pipeline parallelism (PP). In the case of TP, its integration with LASP-2 is straightforward and efficient. Linear attention layers apply TP to break down matrix operations across both intra-chunk and inter-chunk computations. At the same time, the MLP layers are processed as usual under TP, without any modification. When LASP-2 is paired with PP, instead of using traditional micro-batches, it substitutes them with sub-sequences extracted from the mini-batch. One key difference from standard PP is that each device locally and specifically stores the intermediate states, $\mathbf{M}_t$ during the forward pass and $\mathbf{dM}_t$ during the backward pass without communicating these states to other devices.

### A.4.2 VARIABLE LENGTH

During pretraining, the batch typically contains sequences of uniform length. However, when finetuning or during inference, the model might encounter input sequences of varying lengths. A straightforward solution to address this is to right-pad all sequences in a batch to match the length of the longest sequence. Unfortunately, this method can be inefficient, especially when the lengths differ significantly across sequences. For standard transformers, more sophisticated approaches have been developed to handle this challenge. These include techniques like load-balancing across GPUs without padding (Zeng et al., 2022; Zhai et al., 2023) or packing multiple sequences into a single batch and adjusting the attention mask accordingly (Ding et al., 2024; Pouransari et al., 2024).

---

**Algorithm 7** AllGather-based Context Parallelism

---

1: **Input:** input sequence $\mathbf{X}$, distributed world size $W$, sequence parallel size $T = W$.
2: Distribute $\mathbf{X} = [\mathbf{X}_t]_1^T$.
3: **for** chunk $t \in \{1, \cdots, T\}$ on ranks $\{1, \cdots, W\}$ in parallel **do**
4:     Calculate $\mathbf{Q}_t = \mathbf{X}_t \mathbf{W}_Q$, $\mathbf{K}_t = \mathbf{X}_t \mathbf{W}_K$, $\mathbf{V}_t = \mathbf{X}_t \mathbf{W}_V$.
5:     Communicate $[\mathbf{K}_t]_1^T = \text{AllGather}([\mathbf{K}_t]_1^T)$ and $[\mathbf{V}_t]_1^T = \text{AllGather}([\mathbf{V}_t]_1^T)$.
6:     Concatenate $\mathbf{K} = \text{Concat}([\mathbf{K}_t]_1^T)$ and $\mathbf{V} = \text{Concat}([\mathbf{V}_t]_1^T)$.
7:     Compute $\mathbf{O}_t = \text{Softmax}(\mathbf{Q}_t \mathbf{K}^\top / \sqrt{d}) \mathbf{V}$.
8: **end for**
9: **return** $\mathbf{O} = [\mathbf{O}_t]_1^T$.

---

LASP-2 can manage variable sequence lengths efficiently by treating the entire batch as a single long sequence, streamlining the process without requiring padding.

## A.5 ADDITIONAL EXPERIMENT RESULTS

### A.5.1 BIDIRECTIONAL LANGUAGE MODELING TASK

To evaluate on the bidirectional language modeling task, we take RoBERTa as the base model and train on 4 A100 GPUs for 50K iterations with a total input sequence length of 2048. As the results shown in Table 3, LASP-2 with Linear Attention is able to reach an approximate convergence performance with Ring Attention on the standard attention based model.

Table 3: **Convergence Performance on Bidirectional Language Modeling Task.**

| Method | Training Loss | Validation Loss |
|---|---|---|
| Standard Attention (Ring Attention) | 1.815 | 1.957 |
| Basic Linear Attention (LASP-2) | 1.812 | 1.961 |

### A.5.2 ABLATION STUDY ON HYBRID RATIO

We provide ablation results on the hybrid ratio of hybrid models. Let "L" denotes Linear Transformer layers and "N" denotes normal Transformer layers. The hybrid models evaluated here have architectures of: 0 Hybrid: "LLLL LLLL LLLL LLLL"; 1/8 Hybrid: "LLLL LLLN LLLL LLLN"; 1/4 Hybrid: "LLLN LLLN LLLN LLLN"; 1/2 Hybrid: "LNLN LNLN LNLN LNLN". Comparing with the Llama3-1B baseline using standard attention, which has a loss value of 2.759, it is clear that higher hybrid ratios tend to lead better convergence performance, but sometimes, a moderate hybrid ratio may reach a better result.

Table 4: **Ablation Study on Hybrid Ratio in Hybrid Models.** Loss values are reported in the Table.

| Attention Module | 0 Hybrid (Pure Model) | 1/8 Hybrid | 1/4 Hybrid | 1/2 Hybrid |
|---|---|---|---|---|
| Basic Linear Attention | 2.892 | 2.826 | 2.824 | 2.775 |
| GLA | 2.845 | 2.751 | 2.754 | 2.753 |

### A.5.3 ABLATION STUDY ON VARYING SIZES OF GATHERING

We have conducted ablation study on varying sizes of gathering memory states. Considering a batch size of 1, in the Linear-Llama3-1B model, the tensor shape of memory state is [1, 16, 2048, 2048]. We use 64 GPUs and a sequence lenghth of 1024K, repeat each test 10 times and report their mean values. We change the split size of gathering memory states and present the LASP-2 throughput results in Table 5. It can be seen that smaller split size (i.e., more splits) tends to lead lightly slower throughput. The results show that the utilization of all-gather operation is not the only

reason of efficiency enhancement. The communication manner as well as the computational order reorganization also play an important role.

Table 5: **Throughput Results (tokens/sec) on Varying Split Sizes of Gathering.**

| Split Size | 2048 | 512 | 128 | 32 |
|---|---|---|---|---|
| **Number of Splits** | 1 | 4 | 16 | 64 |
| **Throughput** | 486183 | 486166 | 486169 | 486158 |

### A.5.4 QUANTITATIVE SCALABILITY RESULTS

See Table 6 in next page.

Table 6: **Quantitative Scalability Results of LASP-2 on Throughput (tokens/sec) and Memory Usage Per GPU (GB).** Experiments are performed on Linear-Llama3-1B, scaling sequence length from 2K to 4096K with a batch size of 1.

| Sequence Length | Number of GPUs | Throughput | Memory Usage Per GPU |
|---|---|---|---|
| **2K** | 16 | 1254 | 25.6 |
| | 32 | 1209 | 25.6 |
| | 64 | 1285 | 25.6 |
| | 128 | 1205 | 25.6 |
| **4K** | 16 | 2478 | 25.6 |
| | 32 | 2446 | 25.6 |
| | 64 | 2327 | 25.6 |
| | 128 | 2344 | 25.6 |
| **8K** | 16 | 4835 | 25.6 |
| | 32 | 4784 | 25.6 |
| | 64 | 4693 | 25.6 |
| | 128 | 4678 | 25.6 |
| **16K** | 16 | 9530 | 25.6 |
| | 32 | 9494 | 25.6 |
| | 64 | 9305 | 25.6 |
| | 128 | 9313 | 25.6 |
| **32K** | 16 | 18105 | 28.7 |
| | 32 | 17755 | 25.6 |
| | 64 | 17835 | 25.6 |
| | 128 | 17807 | 25.6 |
| **64K** | 16 | 35507 | 33.8 |
| | 32 | 34240 | 28.7 |
| | 64 | 34118 | 25.6 |
| | 128 | 33344 | 25.6 |
| **128K** | 16 | 68406 | 40.2 |
| | 32 | 68545 | 33.8 |
| | 64 | 67344 | 28.7 |
| | 128 | 66811 | 25.6 |
| **256K** | 16 | 135635 | 57.8 |
| | 32 | 132605 | 40.2 |
| | 64 | 130215 | 33.8 |
| | 128 | 131550 | 28.7 |
| **512K** | 16 | OOM | OOM |
| | 32 | 250586 | 57.8 |
| | 64 | 245353 | 40.2 |
| | 128 | 233442 | 33.8 |
| **1024K** | 16 | OOM | OOM |
| | 32 | OOM | OOM |
| | 64 | 442221 | 57.8 |
| | 128 | 416465 | 40.2 |
| **2048K** | 16 | OOM | OOM |
| | 32 | OOM | OOM |
| | 64 | OOM | OOM |
| | 128 | 769030 | 57.8 |
| **4096K** | 16 | OOM | OOM |
| | 32 | OOM | OOM |
| | 64 | OOM | OOM |
| | 128 | OOM | OOM |

