# OpenReview forum: "LASP-2: Rethinking Sequence Parallelism for Linear Attention and its Hybrid"
_ICLR.cc/2025/Conference — Submitted to ICLR 2025_

### Official Review · Reviewer_Zvcj · 2024-11-03

**Soundness:** 3
**Presentation:** 4
**Contribution:** 3
**Rating:** 6
**Confidence:** 4

**Summary:**

This paper considers the problem of efficient parallel implementation of linear attention models. At a _very_ high level, the idea is something that has been explored before-- i.e. just "chunk" up the computation of the "KV cache." The main new insight of the paper (which as far as I can tell is new and very nice!) is that in the forward pass, one can pretty much parallelize the entire computation of the KV cache _except_ for _one_ aggregation step (which is basically summing up the KV cache up until the last step). The paper consider both a linear attention model (where all attention layers in Transformers is replaced by linear attention-- including lot of variants from the original linear attention paper) as well as hybrid model that has both softmax and linear attention layers.

The paper presents bunch of implementations and on SlimPajama corpus the hybrid model with Gated Linear Attention improved both on accuracy and throughput over standard Attention.

**Strengths:**

* The proposed algorithm is simple and uses a nice idea. (The simplicity is good!)

* Even though experimental results are not presented for the H100, from what I understand, the aggregation step mentioned above is natively supported in H100, which could improve things even more.
    * If the authors have access to H100, it would be good to see if this actually happens. _However, I'm not expecting the authors to do this during the rebuttal phase._

* The experimental results show improvement over both existing parallel Linear Attention models as well as standard Attention.

* The paper is well written.

**Weaknesses:**

Overall, I'm reasonably happy with the paper but I think the paper would be stronger with two additions (adding a theoretical analysis and accuracy on other benchmarks) and by addressing bunch of other smaller comments.

Theoretical Analysis
--------------------

While the experiments do show an improvement over LASP (and other parallel implementation like Ringed Attention), I do not have a sense of what is the _theoretically maximum_ gain that could be achieved with LASP-2? I think the paper would be much stronger if there were a theoretical analysis to show what would be the _ideal_ improvement (in the limit of $N\to\infty$) of LASP-2 over LASP and Ringed Attention?

The reason I as this is because I'm not sure if the proposed method's improvement is a constant factor (like $2\times$) or is is super-constant.

I understand that matching the theoretical gains in experiments is hard (and I'm not expecting this) but I want to get a sense of what is the best-case scenario improvement with the proposed method.

Moving beyond one benchmark
---------------------------

The experimental results on performance was only done on the SlimPajama benchmark. It is hard to calibrate how good a system is if comparison is made on only one benchmark. It would make the paper stronger if there were comparisons done on other benchmarks, ideally with different language modeling tasks.

(Relatively) Minor Weaknesses
-----------------------------

All the comments below are more along the lines of asking clarifications on some of the experimental choices made in the paper and the rest are presentation related. I'm listing these in order of when they appear in the paper and not in the order of their importance:

1. [Line 135] This is a nitpick: the $k_i$ and $v_i$ should be $\mathbf{k}_i$ and $\mathbf{v}_i$.
2. [Lines 142-143] Could you please expand on what it means that ignoring the normalizing denominator "works effectively in practice"? My understanding was that the normalization was important even for linear attention models.
3. [Line 377] Combing smaller documents into one seems like would lead to worse results accuracy-wise?
4. [Line 430] Could you please elaborate exactly which parts of Based were used? Specifically Based a Taylor series expansion that approximates Attention as well as an exact sliding window standard Attention. I'm guessing the experiments in the paper used both of these parts of Based but I just wanted to double-check.
5. [Fig 3+4] Do you get similar results when replacing basic linear attention with other linear attention variants considered in the paper?

**Questions:**

Please address (to the extent possible) the weaknesses outlined above.

Post-Rebuttal Comments
------------------------------

The rebuttals answered most of my questions. I encourage the authors to present the theoretical analysis of LASP-2 vs LASP-1 as they did in their rebuttal in the next version of their paper. I'm upping my score to a 6.

---

### Official Review · Reviewer_bdiQ · 2024-11-03

**Soundness:** 2
**Presentation:** 3
**Contribution:** 2
**Rating:** 5
**Confidence:** 3

**Summary:**

This paper deals with the linear attention, which replaces the exponential kernel in softmax attention with a simpler dot product between key and query vectors. To improve the sequence parallelism the authors propose LASP-2, which reorganizes the computation and communication procedure with an optimized execution order.

**Strengths:**

The main strong point are as follows:
- only a single all-gather collective communication is needed in the forward or backward of each iteration.
- communication and computation parallelism is improved significantly as it is shown by experiments.

**Weaknesses:**

I list the main weaknesses below:
- In my opinion, the structure of the paper can be reworked. Namely, the section «Related work» should be in the beginning.
- The differences of LASP-1 and LASP-2 (section 3.3) look rather marginal. Could you please elaborate more on this?
- The proposed method is validated only by experiments on one dataset. There are no theoretical guarantees, thus it is difficult to judge how good the method is. I would suggest either add some theoretical results or extend the experimental part.

**Questions:**

See weaknesses

---

### Official Review · Reviewer_mdwz · 2024-11-04

**Soundness:** 2
**Presentation:** 3
**Contribution:** 2
**Rating:** 6
**Confidence:** 4

**Summary:**

This paper proposes LASP-2, a new sequence parallelism method for linear attention improving the existing method of LASP (Sun et al., 2024).
The authors claim that **using a single all-gather collective communication on memory states for each device improves communication and computation parallelism.** To support their claim, the authors provide an efficient parallelism for integrating output for each chunk where the communication cost is independent of the sequence or chunk length. The authors also show a variant to enforce causal constraints by calculating the attention output as a sum in inter-chunk production and intra-chunk one. LASP-2 is also extended to LASP-2H combining linear transformer layers with standard softmax self-attention to enhance long-context capabilities. Experimental results on variants of Linear-Llama3 models show improvement in the speed and scalability of LASP-2 and LASP-2H.

**Strengths:**

- The concept of linear attention and the proposed single all-together collective communication on memory states are described in detail, improving overall readability.
- This paper empirically supports the effectiveness of their method in speed and scalability.

**Weaknesses:**

Several key parts need to be clarified.

1. A contribution of this paper is extending LASP-2 to a hybrid model that integrates linear and standard attention layers.
However, the description **On Standard Attention Module** on page 7 is unclear. How is the proposed all-gather collective communication used on standard attention that includes softmax attention? It seems that the same communication is not valid since the softmax attention hinders the right-product-first trick. Could the authors explicitly write down equations describing this part?

2. How are the 1/4 hybrid models constructed? Are the first 1/4 layers linear modules? Could the authors provide an ablation study on the ratio (1/4 here) as their claim lies in the performance of the hybrid model?

3. In LASP-2 with masking, there is a discrepancy in attention weights between inter-chunks and intra-chunks. Is there any possibility that the intra-chunks are pronounced more than inter-chunks as the weights in intra-chunks are always positive?

4. The LASP paper claims that LASP scales sequence length up to 4096K on 128 GPUs. Does LASP-2 give better scalability than SASP in that setting?

5. To further emphasize the novelty of LASP-2, I recommend describing LASP and the superior properties of LASP-2 in detail with exact formulations.

**Questions:**

See the above weaknesses part.

---

### Official Review · Reviewer_Npbh · 2024-11-05

**Soundness:** 3
**Presentation:** 3
**Contribution:** 2
**Rating:** 6
**Confidence:** 3

**Summary:**

LASP-2 is proposed as an improvement upon LASP of Sun et al. 2024. Like LASP, LASP-2 implements linear attention based on sequence parallelism, but with better communication and computation efficiency. Central to the proposed idea is a single all-gather communication of the memory state matrix with dimension independent of the sequence or chunk length. The overall cost scales linearly with the number of devices involved in the SP communication group. An extension is also given to handle both linear and standard attention modules. Experiments are conducted using sequences as long as  2048K to validate the proposed methods.

**Strengths:**

The paper is well written with nice tutorial on linear attention.

Practical merits are high.

The allgather operation for making the state information of all chunks available to every device in the system seems unique and appears to be the reason behind the performance advantage observed for very long sequences.

**Weaknesses:**

The claim here is that relative to LASP of Sun et al. 2024, the proposed method, coined LASP-2, offers better communication efficiency since it is the state matrix that gets transferred whose dimension does not depend on the sequence or chunk length, as opposed to K and V matrices that depend on the chunk length. But this is not true. The communication requirement is the same with the original LASP algorithm that communicates the KV activation.

Likewise, the claim above and below eq (5) about concurrent computations for each chunk being unique to LASP-2 is unfounded. It is easy to see that identical operations are already done in LASP.

**Questions:**

Given the above weakness concerns, the only significant difference between LASP and LASP-2 seems to be the allgather operation (and the subsequent update on the aggregated state) in the latter that allows every device in the system to have access to the complete set of memory states for entire chunks.

In light of this, ablation on varying sizes of gathering would be desired to verify whether the operation of (6) (and (8) in the case of masking) is indeed responsible for the improved throughput for length greater than 1024K in Fig. 3.

---

### Meta-Review · Area_Chair_ttih · 2024-12-21

**Metareview:**

This paper proposed LASP-2 as a sequence parallelism (SP) method to improve the communication and computation efficiency for linear attention models. The key idea is to reorganize the computation and introduce an all-gather communication strategy. Additionally, LASP-2H extends LASP-2 to hybrid models combining linear and standard attention layers. The authors validate their approach on a large-scale dataset and conduct several ablation studies.

Strengths:
1. The use of all-gather communication is interesting, which reduces the communication bottlenecks in LASP-1, and thus improves throughput and scalability.

2. The integration of LASP-2 into hybrid models provides one solution for long-sequence tasks that uses a mix of attention mechanisms.



Weaknesses:
1. Reviewers (e.g., bdiQ, mdwz) noted that the improvements over LASP-1 are incremental. Moreover, the main improvement comes from a reorganization of the computation, leading to marginal novelty.


2.  There is a lack of theoretical analysis or justification on the improvement of the proposed method.

3.  The hybrid implementation (e.g., handling of standard attention layers) remained unclear.

4.  The experimental part is insufficient.  More diverse dataset and benchmarks are suggested.


Overall, while this paper is quite interesting,  I have to recommend rejection based on the above main weaknesses raised by the reviewers. I do encourage  the authors to revise it accordingly before acceptance.

**Additional Comments On Reviewer Discussion:**

1. Reviewer mdwz requested some evaluations on diverse tasks to demonstrate the generalizability of the proposed method. The authors provided additional results but did not extend to broader benchmarks.

2. Reviewers bdiQ and mdwz concerned the marginal improvements of LASP-2 over LASP-1. The authors clarified their approach but did not fully address these concerns.

3. Reviewer mdwz raised some questions about the implementation of hybrid model, e.g.,  the handling of standard attention layers. While the authors offered some explanations, certain aspects remained vague.

4. Reviewer Zvcj suggests some theoretical analysis of LASP-2’s potential gains, which the authors did not provide.

---

### Decision · Program_Chairs · 2025-01-22

Reject